# OpenReview forum: "Gandalf the Red: Adaptive Security for LLMs"
_ICML.cc/2025/Conference — ICML 2025 poster_

### Official Review · Reviewer_mvGt · 2025-03-13

**Overall Recommendation:** 3

**Summary:**

The paper introduces a crowdsourced platform (RedCrowd) to test model security defences against prompt attacks and quantify their usability impact. Towards that they propose D-SEC a model for assessing the security-utlilty trade off. Using 279,000 attacks they demonstrate the strengths and weaknesses of various defences.

**Claims And Evidence:**

The claims are partially supported but there are also points with insufficient evidence.

I found the claim that stricter security measures reduce usability, supported. As well as, the defence-in-depth finding.

I found the "adaptiveness" rather simplistic so I'm not sure if we can claim generality in this case. Moreover, I am not convinced that the attacks are realistic enough to draw conclusions about the behaviour/response of the models in high-risk prompts.

**Essential References Not Discussed:**

N/A

**Experimental Designs Or Analyses:**

Please see "Methods And Evaluation Criteria" but I would like to ask why newer models such as 4o (which was available at the time) were not included in the experiments. It is possible that more capable models will exhibit distinctly different trade offs.

**Methods And Evaluation Criteria:**

I found the idea of crowdsourcing data an excellent way of collecting traces from various types of "adversaries" with different approaches and strengths.

However, I have concerns about the validity of the results. In particular:

1. I don't think that the security/utility trade off (formula 1) provides any actionable information. Such metrics have been studied for some time in computer security & usability field and identifying high quality metrics is more nuanced (e.g., Savola, Reijo M. "Quality of security metrics and measurements." Computers & Security 37 (2013): 78-90.)

2. As I mentioned earlier, I don't think the tasks are reliable to extract conclusions about the real capabilities of the models in tested. One of my concerns is sandbagging. In this case, the model may not take the task seriously (e.g., password) and thus does not actually "do it's best". Another (smaller) issue is with assessing the utility of the model as the quality of the responses may degrade without refusal. Cosine similarity and response length may give an indication but working with embeddings would probably give more reliable results.

**Other Comments Or Suggestions:**

N/A

**Other Strengths And Weaknesses:**

I found the paper well written and accessible. It develops and introduces concepts in an intuitive manner, making it easy to follow.

**Questions For Authors:**

1. I wonder whether it'd be possible to replay traces of past attacks on newer models? This would help update the findings without having to involve participants again. While, of course, the exchange will not be tailored, it may still result in successful attacks (or blocks) which would still be informative (but qualitatively different to the live human attacks).

**Relation To Broader Scientific Literature:**

There are only a few works in this space that involve such a high number of participants. This is a strong point for this work as the attackers are considerably more realistic than what's used in many other works.

**Theoretical Claims:**

N/A

---

> ### Author Rebuttal · Authors · 2025-04-01
>
> Thank you for taking the time to review our paper.
>
> We group your comments and questions below and respond to each separately.
>
> **Generalizability of the results**
> - The D-SEC framework is not specific to any one application and generalizes beyond the experimental “password extraction” setup presented in the paper. Practitioners can apply security and utility metrics tailored to their use case.
> While the evaluation focuses on the narrow use case of password extraction, the findings provide generalizable insights as we observe consistent patterns across different target LLMs. For example, selecting a more relevant benign distribution has a consistent and measurable impact on the security-utility tradeoff across models.
> - We appreciate your comment regarding the use of embeddings in measuring utility drop beyond over-refusal. We would like to clarify that our method does use embeddings to compute cosine similarity, as described in Figure 4. [We are happy to make this clearer in the main text if helpful.]
> - To acknowledge the limitations of the task, we will add the following sentence to the Discussion section: “The insights from RedCrowd are limited by the narrow application to password extraction, and a broader empirical analysis is needed to draw application-agnostic conclusions.”
>
> **Actionable insights from the security-utility trade-off**
> - An advantage of Eq 1 is that it can be explicitly optimized, providing practitioners with a principled method for selecting defenses based on their specific goals.
> - In the “Defense in depth” section, we show how optimizing the aggregation of multiple defenses for different $\lambda$ yields a strictly better tradeoff than naive “or” or “and” strategies.
> - Beyond its practical optimization benefits, the framework highlights the often subtle but important ways in which security interventions can affect user utility, something we believe deserves more attention in the field.
>
> **Experiments with newer models**
>
> We limited our game design and analysis to three target LLMs. We selected models based on two criteria: (a) widespread use in real-world applications, and (b) variation in quality and size. GPT-4 was chosen over GPT-4o, which was not yet available when we designed the game protocol. Additionally, we were concerned that a stronger model might make it too hard for players to succeed, leading them to give up early, though we didn’t test this extensively. Expanding to a larger set of models remains an important direction for future work.
>
> We appreciate the suggestion to resubmit prompts to newer models. It is important to note that we cannot replicate our full setup: the adaptive, multi-turn nature of gameplay is lost, and password reveals (central to our success metric) are difficult to detect automatically, especially for obfuscated outputs (see Appendix G). However, two parts of our study can be re-run: the defense-in-depth and utility experiments.
>
> We re-ran both on GPT-4o and will include full figures and tables in the appendix. We provide a summary of findings below.
>
> **Defense-in-depth**: GPT-4o shows a higher “unblocked” rate (see Figure 5), indicating more frequent circumvention of explicit refusals. Many responses avoid giving the password without triggering simple refusal heuristics, reinforcing the need for stronger semantic detection.
>
> **Utility**: The table below reports SCR-derived false positive rates for GPT-4o alongside GPT-4 and GPT-4o-mini, including upper bounds of 95% confidence intervals. GPT-4o is consistently less likely to refuse benign inputs, particularly under strong system prompt defenses (C3).
>
> | Dataset        | Model        | B             | C2            | C3             |
> |----------------|--------------|---------------|---------------|----------------|
> | BasicUser      | GPT-4o-mini  | 0.0% (0.37%)  | 0.3% (0.88%)  |  0.1% (0.56%)  |
> |                | GPT-4        | 0.3% (0.89%)  | 0.5% (1.19%)  |  0.2% (0.73%)  |
> |                | GPT-4o       | 0.2% (0.72%)  | 0.1% (0.56%)  |  0.0% (0.37%)  |
> | BorderlineUser | GPT-4o-mini  | 0.0% (6.06%)  | 1.7% (9.09%)  | 11.9% (22.9%)  |
> |                | GPT-4        | 1.8% (9.39%)  | 3.5% (12.1%)  |  3.5% (12.1%)  |
> |                | GPT-4o       | 0.0% (6.27%)  | 1.8% (9.40%)  |  0.0% (6.27%)  |
>
> We also compare utility metrics based on relative prompt length and cosine similarity to undefended responses. GPT-4o outperforms GPT-4 across all settings except prompt length under the weakest defense (B). As with other models, we observe a drop in utility with stronger defenses, but GPT-4o exhibits smaller degradation overall.
>
> | Model        | Length B | Length C3 | CosSim B | CosSim C3 |
> |--------------|----------|-----------|----------|-----------|
> | GPT-4        | 0.982    | 0.852     | 0.962    | 0.933     |
> | GPT-4o       | 0.960    | 0.892     | 0.967    | 0.953     |

---

### Official Review · Reviewer_L5VF · 2025-03-14

**Overall Recommendation:** 3

**Summary:**

This paper points out that the current defenses against jailbreaking could not block adaptive attacks but impose useability penalties on common users. They propose D-SEC, a threat model which models the attackers and the common users in a session view. They then build a platform called RedCrowd to collect prompt attacks and do analysis.

**Claims And Evidence:**

Yes, the paper provides the detailed statistics to characterize the interplay among attack, defense and utility from 279K prompt attacks on RedCrowd platform.

**Essential References Not Discussed:**

None.

**Ethical Review Concerns:**

I think the authors should address the ethics problem in their article as the RedCrowd platform involves common users to jailbreak commercial LLM applications.

**Ethical Review Flag:**

Flag this paper for an ethics review.

**Ethics Expertise Needed:**

["Responsible Research Practice (e.g., IRB, documentation, research ethics, participant consent)"]

**Experimental Designs Or Analyses:**

Yes, the results are convincing.

**Methods And Evaluation Criteria:**

Yes, the author constructed a crowd-source platform RedCrowd which starts from May 2023 to collect the real-world data against GPT series models.

**Other Comments Or Suggestions:**

None

**Other Strengths And Weaknesses:**

The paper is well-written.

**Questions For Authors:**

- I'd like to know the safety policy that RedCrowd cares about. For example, what types of safety violation are collected? And the statistics.
- What is the privacy and the ethics policy on your platform? Would the jailbreaking topics cause harm to the participants?

=====
Post-Rebuttal: I haven't noted that the RedCrowd mainly experiments with the password scenario. The authors should highlight this point at a more clear position. Besides, I'd like to see how this can be extended to more content safety categories, which is not addressed in this work but is of more practical meaning. This explains why I turn down the score to 3.

**Relation To Broader Scientific Literature:**

Yes, it would provide the community with a good dataset for studing the relation between attack intensity and the safety level.

**Theoretical Claims:**

The threat model is reasonable, which characterizes an attacker who learns from feedback to improve its attack, which is a key reason that current defenses can be broken.

---

> ### Author Rebuttal · Authors · 2025-03-31
>
> Thank you for taking the time to review our paper.
>
> We appreciate the reviewer’s interest in the safety policy used in RedCrowd. We have now clarified our ethics, privacy and safety policy in the manuscript by adding the following to the introduction: “RedCrowd is a white-hat red-teaming system designed to identify vulnerabilities in commercial LLMs before they can be exploited maliciously. It does not promote or expose users to unsafe content. The password extraction use case focuses on identifying security weaknesses, not on generating or disseminating harmful outputs.”
>
> Our focus was narrowly scoped to password extraction, which does not involve the generation of harmful or unsafe content. As such, we did not conduct an explicit analysis of broader safety violations committed by players. A more comprehensive taxonomy and analysis of safety violations is an important direction for future work, particularly as we extend RedCrowd to cover a wider range of misuse scenarios.

---

### Official Review · Reviewer_A3iE · 2025-03-14

**Overall Recommendation:** 4

**Summary:**

- This paper tackles a really important issue in LLM security – how do we stop prompt attacks without making the user experience terrible? I really liked how the authors separate attackers from regular users in their D-SEC model. A lot of past work just looks at how well a defense blocks attacks, but this paper actually cares about how it affects legit users too, which is super important.
- The RedCrowd platform is a really cool idea. Instead of just testing defenses with pre-made benchmarks, they set up a gamified system where real people try to break the model in creative ways. That’s way more realistic than static datasets. Plus, they collect a massive dataset (279k attacks), which makes the analysis feel very solid.
- They make a strong case for adaptive defenses. The idea that security shouldn’t just be "block everything suspicious" but should adjust based on the attack patterns is very convincing. Their experiments show that stacking multiple defenses together (defense-in-depth) and making them adaptive works much better than relying on just one method. This is one of the biggest takeaways from the paper.
- One thing that could be clearer is how well these ideas apply to real-world LLM applications. They run all tests in a controlled setting with their RedCrowd platform, which is great for research, but I wonder how this would work in actual deployed systems. Like, do companies need to build their own version of RedCrowd to get similar results? Or can they just apply these techniques out of the box? Some discussion on practical deployment would be helpful.
- Overall, really solid work with a clear real-world motivation. The focus on balancing security and usability is refreshing, and their dataset + analysis are strong contributions. Some details on real-world applicability could make the work even stronger, but the paper definitely moves the conversation in the right direction.

### Nits and Prior Relevant Work Which has not been cited:

- Relevant to cite for possible adversarial attacks against LLMs https://arxiv.org/abs/2407.14937

**Claims And Evidence:**

- The paper shows that pre-made attack benchmarks often give an overly optimistic view of security because they don’t account for how attackers adapt. Using RedCrowd, they find that non-adaptive defenses fail more often when attackers adjust their strategies across multiple attempts.
- Their experiments measure both attack success rates (AFR) and how often normal users get blocked (SCR). They show that some defenses built inside the LLM (like strong system prompts) don’t just block attacks but also degrade normal responses, even when they aren’t outright refusing user requests.
- The results show that combining multiple defenses (e.g., system prompts + input/output filters + adaptive blocking) is far better than using a single one.

**Essential References Not Discussed:**

The paper could be strengthened by talking about https://arxiv.org/abs/2407.14937, which provides a structured threat model for red-teaming LLMs. It organizes red-teaming attacks into different stages of LLM development and deployment, offering a more systematic way to think about security risks. The paper also covers various defense strategies and practical methods for adversarial testing, which align with the goals of this study. Including insights from this work could help place RedCrowd’s findings within a broader security framework and make the comparisons to existing red-teaming methodologies clearer.

**Experimental Designs Or Analyses:**

- The authors use a large-scale, crowd-sourced red-teaming approach (RedCrowd) to collect adaptive attacks. Instead of using static benchmarks, they let real users try to break the LLM defenses in a gamified setting. This results in a dataset of 279k real-world attacks, making the evaluation more realistic than pre-scripted attack lists.
- They compare different types of defenses: single-layer, multi-layer (defense-in-depth), and adaptive defenses. They test system prompts, input/output filters, and combined approaches to see which strategies work best. They also evaluate adaptive defenses that adjust based on session history.
- Security and usability are evaluated separately using Attacker Failure Rate (AFR) and Session Completion Rate (SCR). AFR measures how often attacks fail, while SCR checks how often normal users are blocked. This dual-metric evaluation helps balance security vs. usability.
- They analyze attack strategies by categorizing them based on techniques used (e.g., direct requests, obfuscation, persuasion, context override). This helps understand what types of attacks are most effective and how defenses should be designed to counter them.
- They also test the impact of restricting LLM application domains (general chatbot vs. specific tasks like summarization or topic-based bots). Results show that more restricted LLM use cases are generally easier to secure.

The experimental design and analysis are generally sound, with a strong empirical approach using real-world adaptive attacks, well-defined evaluation metrics, and a thoughtful comparison of different defense strategies

**Methods And Evaluation Criteria:**

- Instead of relying on pre-made attack benchmarks, they let real users try to bypass security in a gamified setting. This helps them collect adaptive, realistic attacks rather than just one-off static prompts.
- They separate attackers from normal users and analyze how defenses impact both. Their model tracks multiple interactions (not just one-time attacks) to see how attackers adapt over time.
- AFR shows how often defenses successfully block attackers from getting the LLM to reveal sensitive info. SCR tracks how often normal users get blocked or experience degraded responses due to security measures.
- They test whether adding multiple defenses together (defense-in-depth) and making them adaptive improves security without hurting usability too much.

**Other Comments Or Suggestions:**

Recommend the authors to include relevant citations and a few lines about goals and capabilities of actors.

**Other Strengths And Weaknesses:**

In the threat model, the goals and capabilities of actors are missing.

**Questions For Authors:**

None

**Relation To Broader Scientific Literature:**

- Prior work on prompt attacks mostly relies on fixed benchmarks or synthetic attack generation. This paper improves on that by using a dynamic, real-world attack dataset collected via RedCrowd, making it more applicable to real-world security challenges.
- The idea of balancing security and usability is well-known in traditional cybersecurity, but it's less explored in LLM defenses. The paper applies these principles to language model security, similar to how adversarial robustness is studied in image recognition or malware detection.
- Work like OpenAI’s red-teaming efforts and adversarial testing in NLP show that human-in-the-loop security evaluation is necessary. This paper adds to that by structuring red-teaming in a crowd-sourced, gamified manner, creating a large and diverse attack dataset.
- Similar to work on prompt injections, jailbreak attacks, and adversarial NLP techniques, this paper helps categorize how users attempt to break LLM security and which defenses are most effective.

**Theoretical Claims:**

While the paper is largely empirical, it makes some theoretical arguments about security modeling and evaluation that are tested through experiments. They introduce a developer utility function that balances security (AFR) and usability (SCR). The D-SEC framework argues that evaluating LLM defenses requires modeling attackers and legitimate users separately. This is a conceptual claim about how security should be assessed rather than an empirical result.

---

> ### Author Rebuttal · Authors · 2025-03-31
>
> Thank you for taking the time to review our paper.
>
> We have addressed both of your suggested improvements in the manuscript and respond explicitly to the comments below.
>
> **Additional citation and discussion on real-world applications**
> - We will make the implications of RedCrowd for real world deployments clearer by adding the following in the discussion: “While our experiments use RedCrowd in a controlled setting, the D-SEC framework is designed for real-world deployment. Developers can apply the core strategies—domain restriction, defense aggregation, and adaptive defenses—using their own user data, without needing to replicate RedCrowd in full. While large-scale crowdsourcing remains the gold standard for red teaming, it is resource-intensive; advancing highly capable automated attackers is an important direction for making these evaluations more practical at scale.”
>
> **Goals and capabilities of actors**
> - We appreciate the suggestion to incorporate adversary capabilities and goals to the threat model as discussed in [ https://arxiv.org/abs/2407.14937]. We will add the following sentence to the Threat Model section: “Following the framework in [1], we restrict the attacker’s capabilities to sending direct inputs to the target LLM via user messages. While indirect attacks via data (e.g., prompt injections in retrieved content) are also compatible with D-SEC, we focus on the direct setting for simplicity. Attacker goals are broad and application-specific.”

---

### Official Review · Reviewer_4Yn6 · 2025-03-16

**Overall Recommendation:** 4

**Summary:**

- the paper studies LLM prompting attacks (crafting prompts to (adversarially) manipulate model behavior
- the paper provides the following:
  - "D-SEC", a threat model for prompting attacks that:
    - encompasses an attacker, a model user (who wishes to use the system for benign purposes), and the model developer (who wishes to balance the system's utility to user by avoiding over-refusal, while improving its resistance to the attacks).
    - considers the *adaptive* nature in realistic attack & usage scenarios
  - RedCrowd: a deployed platform that gamify the data collection of adversarial prompts and defenses
  - The collected datasets from RedCrowd
- the paper justifies the D-SEC threat model based on the collected data and in-the-wild experimentation, and present useful findings such as
  - the inefficacy of system prompts (alone) as a defense strategy
  - the necessity and effectiveness of "in-depth" defense strategies (where combinations of individual strategies outperform the constituents)

## update after rebuttal

I appreciate the authors' rebuttal and will keep my score and my assessment that the paper should be accepted.

**Claims And Evidence:**

Yes, the claims of the paper are generally well-supported by evidence. Some comments:

- The paper claims that a novel threat model (D-SEC) to account for user utility is necessary, and subsequently provides justification through expeirments.
  - However, a key idea behind security-utility trade-off is *reducing over-refusal* (due to safety training), and this itself is not an entirely new idea; e.g., Anthropic explored techniques to maintain high-quality defenses while reducing over-refusal rates [1. 2].

- The paper claims to provide the collected dataset, which can be found in the appendix. I believe this is a very valuable contribution to the community, specifically for future learning-based defenses.

Refs:
- [1] https://www.anthropic.com/news/constitutional-classifiers
- [2] https://www.anthropic.com/news/claude-3-family

**Essential References Not Discussed:**

None that I am aware of.

**Experimental Designs Or Analyses:**

Overall the experimental design and analyses look reasonable. The experiments of the paper look comprehensive.

**Methods And Evaluation Criteria:**

Overall the proposed methods and evaluation make sense. Since the paper focuses on a conceptual framework (D-SEC threat model), an in-the-wild data collection and gaming platform, and dataset collection, there are less of an "evaluation of a proposed algorithm" in the traditional sense.

**Other Comments Or Suggestions:**

- The paper, in my opinion, can also be (and is arguably better) positioned as a datasets/benchmark paper, as the bulk of the analysis is performed with the in-the-wild RedCrowd platform and the collected session data therein, and the findings (e.g., system prompts are ineffective) can be reported independent of the proposed threat model.
- Consider tightening the use of the words "dynamic" vs "adaptive"; e.g., in L121, "adaptive" is better suited for "Viewing the threat model as dynamic...". In general, the core of the paper seems to focus on "adaptive" attacks (which has roots in meaning from the differential privacy literature, e.g. [1]).
- In section 2, there are several forward references to section 4.1, and it breaks the flow if the reader does not yet know what section 4 is about. Consider providing a synopsis of what that section is about before forward references.


[1] https://differentialprivacy.org/privacy-composition/

**Other Strengths And Weaknesses:**

Strength
- The paper is well-written and easy to follow
- References to prior work is comprehensive

Weaknesses:
- The paper structure can use better clarity; right now the sections are perhaps a bit loosely connected. For example, it will help for the paper the clearly list out what the research questions are for the experiments (if any), and what the key findings are.
- The significance of D-SEC as a novel threat model is debatable. The community is aware that a security-only defense is bad (because it rejects too many benign requests; e.g. [1]).

[1] https://arxiv.org/abs/2311.16119

**Questions For Authors:**

N/A

**Relation To Broader Scientific Literature:**

The paper is related to the jailbreaking literature. The key idea of the paper (threat model to account for more than just attack success) is seen in prior work (e.g. see "Claims And Evidence"), though the key contribution of the paper is novel (an in-the-wild interaction dataset and the platform to collect it).

**Theoretical Claims:**

Overall the paper does not make theoretical claims.

I have one minor question about Eq 1 (security-utility trade-off), which is an abstract formulation: while the general formulation makes sense, why should one expect that the "security" and "utility" to be quantifiable in the "same space" as implied by the equation, and that they have a linear relationship?
  - That is, why should a fixed amount of security (say some constant number like 0.1) be equivalent to a fixed amount of utility? Why not some non-linear relationships?
  - I understand that the authors opt for "attacker failure rate (AFR)" and "session completion rate (SCR)" to quantify security and utility, and because these are numbers between 0 and 1, it makes sense to put them in the same equation. But I'm curious if authors explored other relationships between these quantities.

---

> ### Author Rebuttal · Authors · 2025-03-31
>
> Thank you for taking the time to review our paper.
>
> Below, we respond to all of the concerns and questions in your review. Please let us know if we missed anything or should expand on our responses.
>
> **Comments regarding the significance and novelty of the D-SEC framework**
>
> - We appreciate the reviewer’s observation and agree that prior work on over-refusals is highly relevant. We will incorporate an explicit reference to the suggested papers and add the following sentence to the main text: “User utility has previously been discussed in the context of model over-refusals [1, 2], a key way in which security interventions can degrade model usability. We analyze this effect in depth in the experimental section of this paper.”
> - We would also like to clarify that the notion of utility in D-SEC is broader than over-refusals alone.  D-SEC facilitates simple modifications to the utility function to account for general types of user experience degradation. We propose such example utility metrics in Figure 4, where we measure changes in response content independently of over refusals. We expect these effects to be even more pronounced in agent deployments, where the security layer can meaningfully impact the program’s execution flow.
>
> **Question about Eq (1)**
> - You are correct that for Eq (1) to make sense, both security and utility should be measured in comparable units. While this might appear limiting at first glance, it actually only means that the metrics for security and utility need to be appropriately (nonlinearly) transformed before analyzing the security-utility trade-off. We opted for this representation for two reasons:
> (i) It makes it easier for readers to understand the security-utility tradeoff.
> (ii) It encourages practitioners to select an appropriate transformation of their desired metric for which units have a comparable meaning. That said, as you allude to, it may not always be as easy as with AFR and SCR to transform metrics onto a comparable scale. We extended footnote 3 by adding: “Both $\mathcal{Q}_{\mathcal{M}}$ and $\mathcal{R}_{\mathcal{M}}$ are assumed to have been transformed appropriately so they have comparable units.”
>
> **Suggestions on improving paper structure**
>
> - List research questions and findings in experiments: We agree that making our research questions and key findings more explicit will improve the paper’s clarity. We will revise the introduction to (a) clearly state our goal of using large-scale crowdsourced red teaming to evaluate the security and utility trade-offs of leading commercial LLMs, and (b) connect it to the main experimental findings regarding the three strategies consistently improve the trade-off: restricting the application domain, aggregating multiple defenses in a defense-in-depth manner, and using adaptive defenses.
> - In the current manuscript, we use “dynamic” to describe systems or assumptions that evolve over time (e.g., the threat model) and “adaptive” to describe mechanisms that incorporate past observations to modify behavior (e.g., attacks or defenses that respond to prior outcomes). We double-checked all occurrences and made sure they align with these definitions.
> - Forward references: We agree that introducing later sections early will provide clarity. For example, we will revise the text to include: “Section 4.1 summarizes how utility metrics vary with the choice of benign user distribution and highlights that utility degradation can extend beyond over-refusals.”

---

> > ### Comment · Reviewer_4Yn6 · 2025-04-03
> >
> > I appreciate the authors' rebuttal and will keep my score and the assessment that the paper should be accepted.

---

### Decision · Program_Chairs · 2025-05-01

**Decision:**

Accept (poster)

**Comment:**

This paper separated attackers from regular users  and studied on how we stop prompt attacks without making the user experience terrible? This paper also proposed RedCrowd, a crowd-sourced, gamified red-teaming platform designed to generate realistic, adaptive attack. All reviewers gave very positive scores. AC read the rebuttal and all reviews, and agrees that the paper addresses an important problem in the literature. AC agrees with the reviewers that this paper should be accepted.